# MCMAE: Masked Convolution Meets Masked Autoencoders

**Peng Gao**[1]   **Teli Ma**[1]   **Hongsheng Li**[1,2]   **Ziyi Lin**[2]   **Jifeng Dai**[3]   **Yu Qiao**[1]
[1] Shanghai AI Laboratory, Shanghai, China
[2] MMLab, CUHK    [3] SenseTime Research

## Abstract

Vision Transformers (ViT) become widely-adopted architectures for various vision tasks. Masked auto-encoding [2, 1, 28, 55] for feature pretraining and multi-scale hybrid convolution-transformer architectures [12, 21, 49, 34, 57] can further unleash the potentials of ViT, leading to state-of-the-art performances on image classification, detection and semantic segmentation. In this paper, our MCMAE framework demonstrates that multi-scale hybrid convolution-transformer can learn more discriminative representations via the mask auto-encoding scheme. However, directly using the original masking strategy leads to the heavy computational cost and pretraining-finetuning discrepancy. To tackle the issue, we adopt the masked convolution to prevent information leakage in the convolution blocks. A simple block-wise masking strategy is proposed to ensure computational efficiency. We also propose to more directly supervise the multi-scale features of the encoder to boost multi-scale features. MCMAE-Base improves ImageNet-1K finetuning accuracy by 1.4% compared with MAE-Base. On object detection, MCMAE-Base finetuned for only 25 epochs surpasses MAE-Base fined-tuned for 100 epochs by 2.9% $AP^{\text{box}}$ and 2.2% $AP^{\text{mask}}$ respectively. Code and pretrained models are available at https://github.com/Alpha-VL/ConvMAE.

## 1   Introduction

Self-supervised learning frameworks, such as DINO [6], MOCO-V3 [10], MAE [28], unleash the potential of Vision Transformers (ViT) and achieve high performance on various downstream vision tasks [33, 30, 58]. Among them, Mask Autoencoders (MAE) [28] demonstrate superior learning ability and scalability. Motivated by BERT [15, 46, 4] in natural language processing, MAE utilizes an asymmetric encoder and decoder architecture, in which masked tokens of the encoder are reconstructed by the decoder. Experiments show that MAE can learn discriminative and scalable representations from ImageNet-1K [14] without relying on large-scale datasets, such as ImageNet-22K.

Local inductive bias [49, 21, 34, 12, 19, 57] and hierarchical representations [42, 53] are explored for boosting the performance of ViT. The combination of local convolution and global transformer operations leads to clear improvements on image classification [33], object detection [30], and semantic segmentation [58]. In contrast to MAE [28], well-performing multi-scale backbones built upon local and global operations are mainly trained in supervised manner. A natural question is whether multi-scale backbone with local and global operations, which show promising performance on supervised learning can be exploited to enhance the masked auto-encoding paradigm [28, 15, 2, 65].

In this paper, a simple and effective self-supervised learning framework, dubbed as MCMAE, is proposed to train scalable representations by introducing hybrid convolution-transformer architectures and masked convolution into the masked auto-encoders. Although the modifications to the original MAE are minimal, MCMAE shows great success on pretraining visual representations for boosting the performances of various tasks.

36th Conference on Neural Information Processing Systems (NeurIPS 2022).

Different from MAE [28], the encoder of MCMAE progressively abstracts the input image into multi-scale token embedding, while the decoder reconstructs the pixels corresponding to masked tokens. For high-resolution token embedding at early stages, convolutions blocks are adopted to encode local content. For low-resolution token embedding at late stage, transformer blocks are used to aggregate global context. The encoder therefore obtains both local and global FOV at different stages and generates discriminative multi-scale features. Note that the MCMAE encoder is partly motivated by the strong hybrid convolution and transformer backbones, including Co-AtNet [12], Early Convolution [57], Container [21] and Uniformer [34]. However, previous hybrid convolution-transformer networks were either not explored for masked auto-encoding [21, 34, 20] or show very similar performance to MAE [52, 59]. Instead of designing novel architectures, we focus on making basic hybrid convolution-transformer architectures work for mask auto-encoding and conduct extensive experiments to demonstrate its effectiveness on various downstream tasks.

The efficient and effective training of MCMAE is enabled by a block-wise masking strategy with masked convolution [60, 25, 31, 48, 24, 40]. The masking strategy adopted in current mask-autoencoding frameworks, such as BEiT [2], MAE [28], SimMIM [59], cannot be naively used for MCMAE as all tokens need to be kept in the later transformer stages. This leads to unaffordable computation cost for pretraining large and huge models, losing MAE's efficiency advantage of omitting masked tokens in transformer encoder. In addition, directly pretraining with the convolution-transformer encoder causes pretraing-finetuning discrepancy as only visible tokens are processed during finetuning stages.

To tackle the issues, we focus on designing hybrid convolution-transformer architectures suitable for mask auto-encoding. Specifically, our MCMAE adopts a block-wise masking strategy to first obtain a mask for the late stage in transformer and then progressively upsamples the mask to larger resolutions in early convolutional stages. In this way, tokens processed by late stages can be completely separated into masked tokens and visible tokens and inherit the computation efficiency of MAE. To prevent information leakage, the convolution blocks at early stages are equipped with masked convolutions, which avoid mixing up features of masked and visible regions in late stages to ensue the training effectiveness. Masked convolution has been well explored in sparse feature extraction [25, 48, 24, 60] and image inpainting [40]. It can be naturally integrated into the hybrid convolution-transformer architecture to enable masked auto-encoding.

Our MCMAE can naturally provide multi-scale features for object detection and semantic segmentation, which are required by modern detection [30] and segmentation frameworks [58]. Multi-scale features from the pretrained MCMAE can significantly improve the performances of object detection and semantic segmentation compared with MAE. MCMAE with masked-based autoencoding can even surpass the fully-supervised pretraining of Swin and MViT [42, 36].

In summary, our contributions can be summarized below: (1) We present the strong and efficient self-supervised framework MCMAE, which is easy to implement but show outstanding performances on different tasks. (2) The proposed MCMAE naturally generates hierarchical representations and exhibit promising performances on object detection. (3) MCMAE-Base improves the ImageNet finetuning accuracy by 1.4% compared with MAE-Base. On COCO 2017 with Mask-RCNN, MCMAE-Base achieves 53.2% $AP^{\mathrm{box}}$ and 47.1% $AP^{\mathrm{mask}}$ with a 25-epoch training schedule while MAE-Base attains 50.3% $AP^{\mathrm{box}}$ and 44.9% $AP^{\mathrm{mask}}$ with 100 training epochs. On ADE20K with UperNet, MCMAE-Base surpasses MAE-Base by 3.6 mIoU (48.1% vs. 51.7%).

## 2 Approach

### 2.1 A Brief Revisit of MAE
Masked Autoencoders (MAE) [28] is a self-supervised method for pretraining ViT by reconstructing masked RGB patches from visible patches. Although MAE has a simple design, it has been proven to be a strong and scalable pretraining framework for learning visual presentations. MAE consists of transformer-based encoder and decoder, where only visible patches are fed into the encoder and learnable mask tokens are processed by the decoder for image reconstruction to learn visual representations. As the encoder only needs to process a small portion of visible tokens, it alleviates the scalability problem to pretrain large vision models.

### 2.2 MCMAE
MCMAE is a simple and effective derivative of the popular MAE [28] with minimal but effective modifications on the encoder design and the masking strategy. The goal of MCMAE is to learn

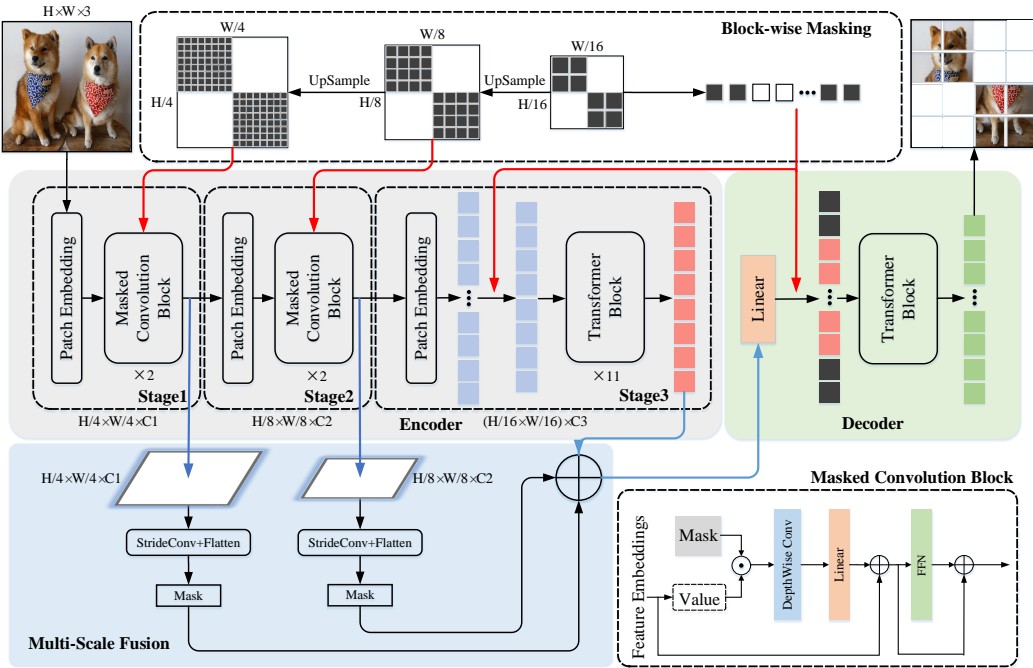

Figure 1: The pipeline of our proposed MCMAE which consists of a hybrid convolution-transformer encoder, block-wise masking strategy with masked convolution and multi-scale decoder.

discriminative multi-scale visual representations and to prevent pretraining-finetuning discrepancy when applies MAE [28] on convolution-transformer networks.

Directly applying the original masking strategy on the feature maps of the convolution-transformer encoder would make transformer layers keeping all tokens during the pretraining, jeopardizing the training efficiency. We introduce a hierarchical masking strategy coupled with masked convolution for the convolution stages to ensure only a small number of visible tokens are input into the transformer layers. The overall pipeline of MCMAE is shown in Figure 1.

**The Hybrid Convolution-transformer Encoder.** There are previous strong hybrid convolution-transformer architectures, such as Co-AtNet [12], Container [21], BoTNet [49], Uniformer [34] and Early Conv [57]. Without using such complicated architectures, we show that a simple design of multi-scale convolution-transformer encoder can already learn powerful representations for various downstream tasks. As shown in Figure 1, our encoder consists of 3 stages with output spatial resolutions of $\frac{H}{4} \times \frac{W}{4}, \frac{H}{8} \times \frac{W}{8}, \frac{H}{16} \times \frac{W}{16}$, respectively, where $H \times W$ is the input image resolution. The first two convolutional stages use convolution blocks to transform the inputs to token embeddings $E_1 \in \mathbb{R}^{\frac{H}{4} \times \frac{W}{4} \times C_1}$ and $E_2 \in \mathbb{R}^{\frac{H}{8} \times \frac{W}{8} \times C_2}$. Our convolution blocks follow the design principle of the transformer block by only replacing the self-attention operation with the $5 \times 5$ depthwise convolution The third transformer stage uses commonly used self-attention blocks to obtain token embeddings $E_3 \in \mathbb{R}^{\frac{H}{16} \times \frac{W}{16} \times C_3}$. Between every stage, stride-2 convolutions are used to downsample the tokens to half of its previous spatial resolution. The local convolutions in stages 1 and 2 have relatively small field-of-view, the transformer blocks in stage 3 aggregate and fuse features from the coarse-grained features and extend the field of view to the whole image. Different from other ViTs, such as CPT [11], Container [21], Uniformer [34], CMT [26], Swin [42], which replace absolute position embedding [42] with relative position embedding or zero-padded convolution at the inputs of the first stage [11, 21, 34, 26], we find that adding absolute position embeddings to the inputs of transformer stage-3 leads to the optimal performance. The class token is also removed from our encoder which shows limited influence.

**Block-wise Masking with Masked Convolutions.** Mask auto-encoders, such as MAE [28] and BEiT [2], adopt a random mask on the input tokens. However, the same strategy cannot be directly applied to our MCMAE encoder. Uniformly masking stage-1 input tokens from the $\frac{H}{4} \times \frac{W}{4}$ feature maps would cause all tokens of stage-3 to have partially visible information and requires keeping all stage-3 tokens. Therefore, we propose to first generate the random mask to mask out $p\%$ (e.g., 75%)

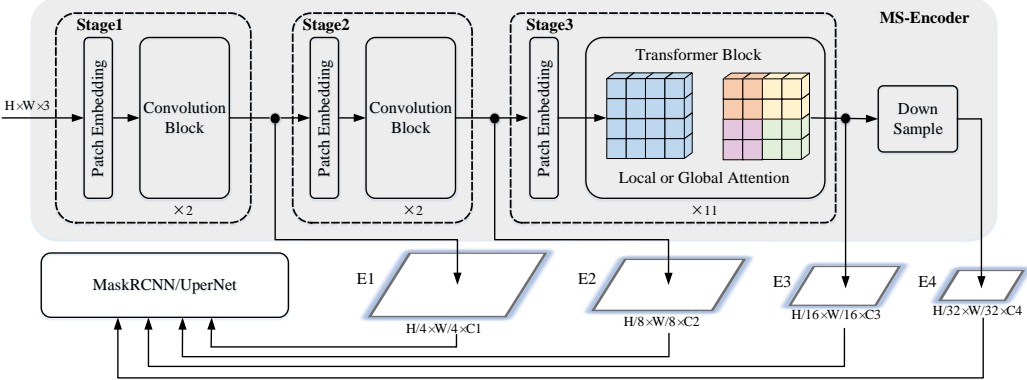

Figure 2: Overview of finetuning MCMAE for object detection and semantic segmentation. The intermediate features of different stages serve as multi-scale inputs for an FPN [38] module.

of stage-3 input tokens and upsample the mask by 2 times and 4 times to obtain the corresponding block-wise masks for masking stage-2 and stage-1 inputs, respectively. The corresponding masked tokens in the three stages are dropped in the encoding process and are reconstructed by the decoder for feature learning. In this way, MCMAE only needs to keep as few as 25% tokens in the time-consuming transformer blocks for training and the efficiency of MCMAE is not compromised.

However, the $5 \times 5$ depthwise convolutions in the first two stages naturally lead to receptive fields larger than the masked patches and cause information leakage when reconstructing masked tokens. To avoid such information leakage and ensure the quality of pretraining, we adopt masked convolution [25, 48] in the first two stages, so that the masked regions would never be involved in the encoding process. The use of masked convolution is crucial to the superior performance of MCMAE and the pretraining-testing discrepancy is prevented by removing partially masked tokens from stage.

**The Multi-scale Decoder and Loss.** The decoder of the original MAE [28] takes as input both visible tokens $E_d$ from the encoder and the mask tokens [Mask], and transform them in stacked transformer blocks for image reconstruction. Our MCMAE encoder obtains multi-scale features $E_1$, $E_2$, $E_3$, captures both fine- and coarse-grained image information. To better supervise the pretraining of such multi-grained representations, we downsample $E_1$ and $E_2$ to the same size of $E_3$ with stride-4 and stride-2 convolutions and fuse multi-grained tokens via a linear layer to obtain visible tokens $E_d$ ,

$$E_d = \text{Linear}(\text{StrideConv}(E_1, 4) + \text{StrideConv}(E_2, 2) + E_3), \tag{1}$$

where $\text{StrideConv}(\cdot, k)$ represents stride-$k$ convolution. The multi-scale decoder is illustrated in the bottom-left part of Figure 1. The same losses from MAE [28] are used for reconstructing masked image patches and only the reconstruction of masked patches are considered in the objective function.

## 2.3 MCMAE for Object Detection and Semantic Segmentation

After pretraining, the proposed MCMAE can naturally generate multi-scale feature maps, which can be processed by existing object detection and semantic segmentation heads.

As shown in Figure 2, to finetune MCMAE for object detection, an $E_4$ feature map of $1/32$ input resolution is first obtained by $2 \times 2$ max pooling $E_3$. However, as the MCMAE stage-3 has 11 global self-attention layers (in our MCMAE-base model) with excessive computational cost, we follow Benchmarking ViT [37] to replace all but 1st, 4th, 7th, 11th global self-attention layers in stage-3 to shifted-window local self-attention layers [42] with alternatively shifted $7 \times 7$ windows. The modified local self-attention layers are still initialized by the pretrained global self-attention layers. A global relative position bias [2, 42, 28, 37] is shared between global transformer blocks. Similarly, a local relative position bias [2, 42, 28, 37] is shared by local transformer blocks. In this way, the heavy computational and GPU memory costs of the stage-3 are much mitigated. The multi-scale features $E_1, E_2, E_3, E_4$ are then fed into the MaskRCNN [30] head for object detection. To finetune MCMAE for semantic segmentation, its stage-3 architecture is kept as the images in segmentation datasets have relatively smaller resolutions. The multi-scale features are feed into UperNet [58].

### 2.4 MCMAE for Video Understanding

Attention based models [54, 63, 50, 3, 42] have demonstrated superior performance on video understanding. Our MCMAE can also be extended to serve as a strong video pretraining framework, dubbbed as VideoMCMAE, with simple modifications. Specifically, VideoMCMAE replaces image patch embedding with cube embedding, after which stage 1 and stage 2 perform local spatial-temporal feature fusion with masked 3D convolutions. Stage 3 still adopts stacked transformer blocks for spatial-temporal fusion. The spatial position embedding is extended to spatial-temporal embedding. Similar to the multi-scale decoder proposed in Section 2.2, outputs from stages 1, 2 and 3 are fused before feeding into a spatial-temporal transformer decoder for masked pixel reconstruction. Details about VideoMCMAE pretraining are in appendix B. Note that unlike previous approaches, which initialize models pretrained on images [34, 36, 7], our VideoMCMAE is pretrained from scratch on pure video datasets.

## 3 Experiments

To validate our proposed MCMAE, we conduct experiments of image classification on ImageNet-1K [14] dataset. The pretrained MCMAE is also extensively tested on object detection and semantic segmentation. By default, we report performance of our the MCMAE-base model *with multi-scale decoder*, which has similar parameters and FLOPs as the MAE-base.

### 3.1 ImageNet-1K Pretraining and Finetuning

**Experimental Setup.** ImageNet-1K [14] consists of 1.3M images of 1k categories for image classification and is split to the training and validation sets. We pretrain our MCMAE on ImageNet-1K training set. By default, we fix the mask ratio to 25% following the original MAE [28]. The decoder is designed to have 8 transformer layers with 512 feature dimensions and 12 attention heads. We adopt a 1600-epoch cosine learning rate schedule with the first 40 epochs for warming up. The AdamW optimizer is utilized with a base learning rate of $1.5 \times 10^{-4}$, a weight decay of 0.05 and a batch size of 1024. Random cropping is employed as data augmentation during pretraining. After pretraining, the MCMAE encoder is used for supervised finetuning on ImageNet-1K training set for 100 epochs using the cosine learning rate schedule. We follow the default finetuning parameters of the original MAE [28] except for the layer-wise learning-rate decay parameters (0.65, 0.75, 0.85). For finetuning, we report the classification accuracy on the ImageNet validation set of the finetuned and pretrained (linear probe) MCMAE encoders.

**Results on ImageNet-1K Finetuning.** We report the accuracy of MCMAE on Table 1 and conduct

| Methods | Backbone | Params. (M) | Supervision | Encoder | P-Epochs | FT (%) | LIN (%) |
|---------|----------|-------------|-------------|---------|----------|--------|---------|
| BEiT [2] | ViT-B | 88 | DALLE | 100% | 300 | 83.0 | 37.6 |
| MAE [28] | ViT-B | 88 | RGB | 25% | 1600 | 83.6 | 67.8 |
| SimMIM [59] | Swin-B | 88 | RGB | 100% | 800 | 84.0 | N/A |
| MaskFeat [55] | ViT-B | 88 | HOG | 100% | 300 | 83.6 | N/A |
| data2vec [1] | ViT-B | 88 | Momentum | 100% | 800 | 84.2 | N/A |
| MCMAE | CViT-B | 88 | RGB | 25% | 1600 | 85.0 | 70.9 |

Table 1: Comparison with state-of-the art mask auto-encoding schemes with similar model size. FT and LIN denotes ImageNet-1K finetuning and linear probe accuracy respectively.

comparisons with state-of-the-art mask autoencoding methods. BEiT [2] pretrains ViT-B through the prediction of visual tokens tokenized by the DALL-E encoder. With 300-epoch pretraining, BEiT can reach a finetuning accuracy of 83.0% and a linear-probe accuracy of 37.6%. Compared with BEiT, MCMAE processes only 25% visible tokens in the encoder and has a lightweight decoder for reconstruction. MCMAE can surpass its finetuning accuracy and linear-probe accuracy by large margins (+2.0%/+33.3%). Compared with the original MAE pretrained for 1,600 epochs, our MCMAE surpasses its finetuning accuracy by 1.4% with same number of pretraining epochs. SimMIM [59] adopts a Swin-B [42] to generate hierarchical representations. MCMAE achieves improvement over its finetuning accuracy (+1.0%). MaskFeat [55] uses HOG [13] features as prediction targets. Data2vec [1] incorporates a momentum encoder [29] to generate predictions in an online manner. Both MaskFeat and Data2vec have higher computational costs than our MCMAE. They can be considered as complementary directions for improving the mask auto-encoding scheme.

### 3.2 Object Detection

**Experimental Setup.** COCO dataset [39] has been widely adopted for benchmarking object detection frameworks. Mask-RCNN [30] is one of the most popular frameworks for object detection. We

| Methods | Pretraining | P-Epochs | F-Epochs | $AP^{box}$ | $AP^{mask}$ | Params (M) | FLOPs (T) |
|---|---|---|---|---|---|---|---|
| Benmarking [37] | IN1K w/o labels | 1600 | 100 | 50.3 | 44.9 | 118 | 0.9 |
| ViTDet [35] | IN1K w/o labels | 1600 | 100 | 51.2 | 45.5 | 111 | 0.8 |
| MIMDET [20] | IN1K w/o labels | 1600 | 36 | 51.5 | 46.0 | 127 | 1.1 |
| Swin+ [42] | IN1K w/ labels | 300 | 36 | 49.2 | 43.5 | 107 | 0.7 |
| MViTv2 [36] | IN1K w/ labels | 300 | 36 | 51.0 | 45.7 | 71 | 0.6 |
| MCMAE | IN1K w/o labels | 1600 | 25 | 53.2 | 47.1 | 104 | 0.9 |

Table 2: Performances of different pretrained backbones on object detection with Mask-RCNN [30].

| Models | Pretrain Data | P-Epochs | mIoU | Params (M) | FLOPs (T) |
|---|---|---|---|---|---|
| DeiT-B [51] | IN1K w/ labels | 300 | 45.6 | 163 | 0.6 |
| Swin-B [42] | IN1K w/ labels | 300 | 48.1 | 121 | 0.3 |
| MoCo V3 [29] | IN1K | 300 | 47.3 | 163 | 0.6 |
| DINO [6] | IN1K | 400 | 47.2 | 163 | 0.6 |
| BEiT [2] | IN1K+DALLE | 1600 | 47.1 | 163 | 0.6 |
| PeCo [17] | IN1K | 300 | 46.7 | 163 | 0.6 |
| CAE [9] | IN1K+DALLE | 800 | 48.8 | 163 | 0.6 |
| MAE [28] | IN1K | 1600 | 48.1 | 163 | 0.6 |
| MCMAE | IN1K | 1600 | 51.7 | 153 | 0.6 |

Table 3: Comparison with different pretrained backbones on ADE20k with UperNet.

employ the encoder of pretraied MCMAE as the backbone for Mask-RCNN. We finetune Mask-RCNN on COCO train2017 split and report $AP^{box}$ and $AP^{mask}$ on val2017 split. We follow most setups of Benchmarking ViT [37]. We report the model performance on object detection under a 25 epochs cosine schedule with a base learning rate of $8.0 \times 10^{-5}$, a weight decay of 0.1.

**Results on COCO 2017.** We compare the performances of state-of-the-art visual backbones in Table 2. Benchmarking ViT [37] extensively explores using plain ViT with Mask-RCNN. Compared with Benchmarkin ViT [37] finetuned for 100 epochs on COCO, MCMAE can significantly improve $AP^{box}$ and $AP^{mask}$ by 2.9% and 2.2% with 25 finetuning epochs. ViTDet [35] improves Benchmarking ViT [37] by introducing a simple feature pyramid module. MIMDet [20] adds a randomly initialized convolution stem and randomly drops input tokens to increase the training efficiency of Mask-RCNN [30]. Note that MIMDet [20] introduces extra parameters due to the incorporation of MAE decoder. Compared with improved version of Benchmarking ViT [37], such as ViTDet [35] and MIMDet [20], MCMAE surpass them by 2.0% and 1.7% with a shorter finetuning schedule (25 epochs vs 100/36 epochs), fewer parameters (104M vs 111M/127M) and similar FLOPs (0.9T). This validates the effectiveness of our proposed MCMAE framework. Swin [42] and MViTv2 [36] are state-of-the-art hierarchical visual backbones. Although adopting a simpler multi-stage architecture, MCMAE outperforms Swin annd MViTv2 by 4.0%/3.6% and 2.2%/1.4% in terms of $AP^{box}/AP^{mask}$. Note that Swin [42] and MViT [36] v2 are pretrained for 300 epochs with 100% tokens in a supervised manner while MCMAE is only pretrained using masked autoencoder with 25% visible tokens, which is efficient for object detection.

### 3.3 Semantic Segmentation

**Experimental Setup.** ADE20K [64] is a widely-used semantic segmentation dataset which contains 25,562 images of 150 fine-grained categories. The dataset is split into training, validation, and testing sets. We leverage the UperNet [58], a hierarchical segmentation network head to compare MCMAE with other backbones. Our MCMAE with UperNet [58] is finetuned on ADE20K training set and tested on validation split. In the training phase, the backbone is initialized with the weights pretrained for 1600 epochs on ImageNet-1K and other modules are initialized with Xavier initialization. We adopt a 16k-iteration polynomial learning rate schedule with the first 1500 iterations for warming up. The AdamW [44] optimizer is adopted with an initial learning rate of $10^{-4}$, a weight decay of 0.05 and a batch size of 16. We follow the default finetuning configurations of MAE on ADE20K except for the feature dimensions for the decoder head and the layer-wise learning rate decay is set as 0.75.

**Results on ADE-20K.** We report the Mean Intersection over Union (mIoU) performance of MCMAE and other state-of-the-art backbones in Table 3. With the 300-epoch pretraining, MoCo V3 [10] can reach 47.2 mIoU when finetuned on semantic segmentation. BEiT [2], PeCo [17] and CAE [9] utilize discrete VAE as visual tokenizer to create the targets. Both BEiT and CAE adopt the DALL-E [47] codebook trained on 250M images, while PeCo trains a codebook only on ImageNet-

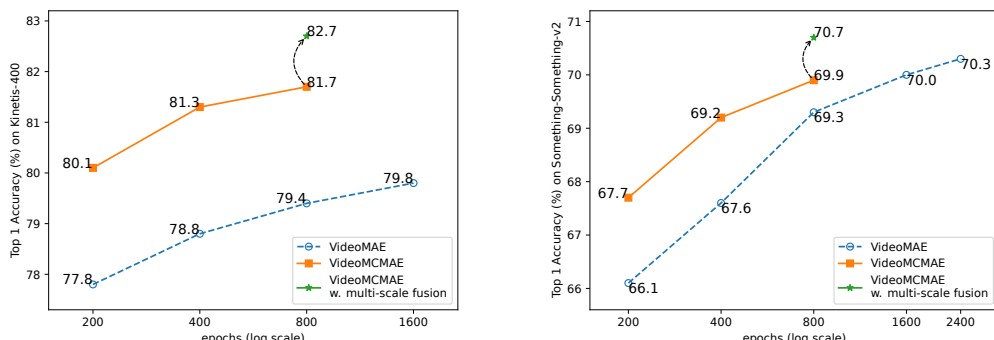

Figure 3: Finetuning accuracy on Kinetics-400 and Something-Something-v2.

1K. Compared with these methods, our 1600-epoch pretrained MCMAE achieves much higher performance (51.7%). Compared with MAE pretrained 1600 epochs, our MCMAE outperforms it by 3.6% mIoU, demonstrating the hierarchical representations of MCMAE largely diminishes the transfer gap between pretrained backbones and downstream networks.

## 3.4 Video Understanding

**Experimental Setup.** To validate the video understanding ability of VideoMCMAE, we pretrain on Kinetics-400 (K400) [32] and Something-something V2 (SSV2) [23] independently and report the finetuning accuracy on K400 and SSV2. The video pretraining and finetuning protocol closely follow the image protocol explained in Section 3.1. For SSV2, we finetune 50 epochs and turn off the flip augmentation. Unlike random masking in image pretraining, tube masking [50] with 90% mask ratio proposed by VideoMAE is adopted as the default masking strategy. For testing, we use the same number of views as VideoMAE for fair comparison, *i.e.*, 3 spatial × 5 temporal views for Kinetics-400 and 3 spatial × 2 temporal views for Something-Something-v2. All results are reported using only the finetuning dataset without extra image or video data.

**Results on K400 and SSV2.** We compare the finetuned accuracy on K400 and SSV2 with Video-MAE [50] for different pretraining epochs. As shown in Figure 3, VideoMCMAE outperforms VideoMAE by a clear margin at 200 and 800 pretraining epochs. Notably, on Kinetics-400, VideoM-CMAE pretrained for 200 epochs slightly outperforms VideoMAE at 1600 epochs, and 800-epoch pretrained VideoMCMAE with multi-scale decoder outperforms VideoMAE at 1600 epochs by more than 2.9%. On Something-Something-v2, our 800-epoch model with multi-scale decoder slightly outperforms VideoMAE at 2400 epochs, which indicates 3x reduction in pretraining epochs.

| Pretrain Epochs | ImageNet | | COCO | | ADE20K |
|---|---|---|---|---|---|
| | FT | LIN | $AP^{box}$ | $AP^{mask}$ | mIoU |
| 200 | 84.1 | 62.5 | 50.2 | 44.8 | 48.1 |
| 400 | 84.4 | 66.9 | 51.4 | 45.7 | 49.5 |
| 800 | 84.6 | 68.4 | 52.0 | 46.3 | 50.2 |
| 1600 | 84.6 | 69.4 | 52.5 | 46.5 | 50.7 |

Table 4: The influence of increasing pretraining epochs on various downstream tasks.

## 3.5 Ablation Study of MCMAE

We conduct extensive ablation studies on MCMAE to analyze different components of MCMAE (see Table 5 and 6). By default, we report the performance of MCMAE *without multi-scale decoder* during ablation studies.

**Pretraining epochs**. For MAE, longer pretraining epochs can significantly improve the learned representations learned. We pretrain MCMAE-Base with 200, 400, 800 and 1600 epochs to test the influences on MCMAE. We report the ImageNet-1K finetuning (FT) and linear probe (LIN) accuracies, $AP^{box}$ and $AP^{mask}$ of COCO, mIoU of ADE20K on Table 4. We observe improved performances on most downstream tasks with longer pretraining epochs.

**Input token random masking.** As shown in Table 5, we replace the proposed block-wise mask strategy with MAE's input token random masking. Compared with our MCMAE-base, the ImageNet-1K finetuning accuracy drops from 84.6% to 84.2% which validates that the proposed simple

block-wise masking strategy can alleviate pretraining-finetuning discrepancy. Input-token random masking results in all tokens in stage-3 being processed by computationally intensive transformer blocks and causes FLOPs to increase by $1.7\times$.

**Influence of masked convolution.** Masked Convolution can prevent information leakage due to the overlapping window in convolution. Removing masked convolution decreases the ImageNet-1K finetuning accuracy from 84.6% to 81.5% , which demonstrates that information leakage in convolution stages hinders feature learning in mask autoencoding.

**Convolution kernel sizes in stages 1 and 2.** Enlarging the kernel size in convolution is shown to be effective for semantic segmentation [45] and visual backbone designs [16, 43]. We also test with enlarging the $5 \times 5$ kernel size in stages 1 and 2 to $7 \times 7$ and $9 \times 9$. As shown by Table 5, we observe that larger kernel sizes barely influence the performance of MCMAE on ImageNet-1K accuracy. We hypothesize that the transformer blocks in stage-3 already provide a global FOV which can cancel out the gains introduced from large kernels.

**Multi-scale Decoder.** In Table 6, we incorporate multi-scale decoder into MCMAE-base and pretrain for 200 and 1600 epochs. Comparared with MCMAE pretrained 200 epochs, multi-scale decoder can improve classification accuracy, detection $AP^{\text{box}}$, detection $AP^{\text{mask}}$ and segmentation mIoU by 0.3%, 0.6% 0.6% and 0.4%, respectively. Given longer pretraining, multi-scale decoder can improve classification accuracy, linear probe accuracy, detection $AP^{\text{box}}$, detection $AP^{\text{mask}}$ and segmentation mIoU by 0.4%, 1.6%, 0.7%, 0.6%, 1.0%, respectively. This indicates that fusing multi-grained tokens for mask reconstruction can lead to powerful representations. We will explore more advanced multi-scale decoder architectures such as UNet in the future.

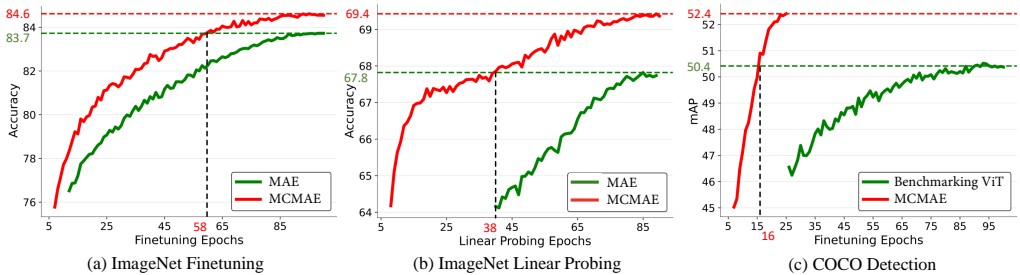

(a) ImageNet Finetuning      (b) ImageNet Linear Probing      (c) COCO Detection

Figure 4: Convergence of MAE and MCMAE on various tasks.

| P-Epochs | Masked Conv | Block Masking | $5 \times 5$ Conv | $7 \times 7$ Conv | $9 \times 9$ Conv | FT (%) | FLOPs |
|---|---|---|---|---|---|---|---|
| | ✓ | ✓ | ✓ | ✗ | ✗ | 84.6 | $1\times$ |
| | ✓ | ✗ | ✓ | ✗ | ✗ | 84.2 | $1.7\times$ |
| 800 | ✗ | ✓ | ✓ | ✗ | ✗ | 81.5 | $1\times$ |
| | ✓ | ✓ | ✓ | ✗ | ✗ | 84.5 | $0.997\times$ |
| | ✓ | ✓ | ✗ | ✓ | ✗ | 84.4 | $1.003\times$ |
| | ✓ | ✓ | ✗ | ✗ | ✓ | 84.6 | $1.007\times$ |

Table 5: Ablation study on the influence of the masked conv, block masking, kernel size in stages 1 and 2 of MCMAE on ImageNet-1K finetuning accuracy.

| P-Epochs | Method | FT (%) | LIN (%) | $AP^{box}$ | $AP^{mask}$ | mIoU |
|---|---|---|---|---|---|---|
| 200 | MCMAE-Base | 84.1 | N/A | 50.2 | 44.8 | 48.1 |
| | w/ multi-scale decoder | 84.4 | N/A | 50.8 | 45.4 | 48.5 |
| 1600 | MCMAE-Base | 84.6 | 69.4 | 52.5 | 46.5 | 50.7 |
| | w/ multi-scale decoder | 85.0 | 70.9 | 53.2 | 47.1 | 51.7 |

Table 6: For a base MCMAE pretrained for 200 epochs and 1600 epochs, we ablate the multi-scale decoder on ImageNet finetuning and object detection on COCO.

**Convergence speed.** We compare the convergence of MCMAE and MAE in terms of ImageNet-1K finetuning, linear probing accuracy and COCO $AP^{box}$ in Figure 4. For fair comparison, MCMAE

and MAE are both pretrained for 1600 epochs. MCMAE not only attains strong final results but also significantly increases convergence speed on various tasks. Specifically, MCMAE can surpass the final performance of MAE at 58 epochs on ImageNet-1K finetuning. On COCO object detection, MCMAE surpasses MAE at 16 epochs, indicating $6.6\times$ faster convergence speed.

## 4   Related Work

**Vision Transformer.** Vision Transformer(ViT) [18, 5] achieved state-of-the-art results on various vision tasks. To increase the convergence speed and improve accuracy, well-explored locality inductive bias have been reintroduced into vision transformer [66, 22, 62, 41, 27, 61, 51, 19, 56, 26], among which, hybrid architecture of convolution and transformer design [49, 57, 12, 21, 34] can achieve state-of-the-art performance of a wide range of tasks. Our MCMAE is highly motivated by the hybrid architecture design [21, 34, 12, 57] in vision backbones. Instead of designing new architectures, MCMAE aim to unleash the powerful representation induced by hybrid architectures through MAE-style pretraining with several insightful modifications.

**Self-supervised Representation Learning.** Contrastive learning [8, 29, 6, 10] learn invariances by comparing augmented views of un-labeled images. Recently Mask-Autoencoding motivated by BERT [15] raised to be a promising methodology. Mask-Autoencoding can learn strong representation through masked patch reconstruction with simple data augmentation. BEiT [2] introduced Mask-Autoencoding into Vision Community. MAE [28] introduced an asymmetric encoder and decoder architecture where masked tokens is skipped in computation-heavy encoder and only pass all tokens through a light-weight decoder. iBoT [65] and Data2Vec [1], PeCo [17] and MaskFeat [55] explore different reconstruction targets. Different from previous improvements of Mask-autoencoding, MCMAE introduce hierarchical representations architectures into MAE.

## 5   Conclusion

We propose a simple self-supervised learning framework named as MCMAE which demonstrate the hybrid local-global blocks [21, 34, 26, 19, 57, 49] can boost the performance of MAE [28] to generate discriminative multi-scale features [38, 53, 42]. The computational efficiency and low pretraing-fineuning gap of original MAE can be well maintained under our MCMAE. MCMAE exhibits significantly improved performances on various vision tasks and can be easily implemented. We will study combining improved reconstruction targets with MCMAE in the future.

**Negative societal impact**: We do not foresee nagative social impact from the proposed work.

**Acknowledgement** : This work was supported in part by the National Natural Science Foundation of China (Grant No. 62206272) and Shanghai Committee of Science and Technology (Grant No. 21DZ1100100).

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

## 6  Appendix

**Architecture Details of MCMAE Encoder.** The details of our hybrid convolution-transformer encoder is explained below. Given an input image $I \in \mathbb{R}^{3 \times H \times W}$, stage 1 of MCMAE encoder generates a high-resolution token embeddings $E_1 \in \mathbb{R}^{C_1 \times \frac{H}{4} \times \frac{W}{4}}$ using non-overlapping $4 \times 4$ strided convolution firstly. Then $E_1$ is feed into stacked convolutional blocks which is repeated $L_1$ times, where $L_1$ stands for the number of layers in stage 1. Similar as stage 1, stage 2 further downsamples feature map into token embeddings $E_2 \in \mathbb{R}^{C_2 \times \frac{H}{8} \times \frac{W}{8}}$ using non-overlapping $2 \times 2$ strided convolution. $E2$ is processed by $L2$ layers of convolutional blocks again. After local information fusion utilized in stage 1 and stage 2, stage 3 perform global feature fusion using transformer block. $E_2$ is projected into tokens embeddings $E_3 \in \mathbb{R}^{(\frac{H}{16} \times \frac{W}{16}) \times C_3}$ using non-overlapping $2 \times 2$ strided convolution. $E_3$ mixing with Intermediate Positional Embedding (IPL) is feed into a pure transformer block with $L_3$ layers. We denote the number of attention heads in stage 3 as $H_a$. The mlp-ratios in FFN for different stages is denoted as $P_1, P_2$ and $P_3$ in respectively. Stage 1 and stage 2 is designed to capture fine-grained details on high resolution feature map. Stage 3 can perform dynamically global reasoning efficiently on a rather low-resolution feature map. At the same time, stage 3 can enlarge the filed-of-view (FOV) of backbone which benefits a wide range of downstream tasks. The encoder of MCMAE can seamlessly inherits the merits of convolution and transformer block. The architecture details for small, base and large model is listed in Table 7. MCMAE small, base, large and huge share similar parameter scale with the encoder of MAE-small, MAE-base, MAE-large and MAE-huge.

| Model | $[C_1, C_2, C_3]$ | $[L_1, L_2, L_3]$ | $[E_1, E_2, E_3]$ | $[P_1, P_2, P_3]$ | $H_a$ | #Params (M) |
|---|---|---|---|---|---|---|
| MCMAE-S | [128, 256, 384] | [2, 2, 11] | [56, 28, 14] | [4, 4, 4] | 6 | 22 |
| MCMAE-B | [256, 384, 768] | [2, 2, 11] | [56, 28,14] | [4, 4, 4] | 12 | 84 |
| MCMAE-B* | [256, 384, 768] | [2, 2, 11] | [56, 28,14] | [8, 8, 4] | 12 | 88 |
| MCMAE-L | [384, 768, 1024] | [2, 2, 23] | [56, 28, 14] | [8, 8, 4] | 16 | 322 |
| MCMAE-H | [768, 1024, 1280] | [2, 2, 31] | [56, 28, 14] | [8, 8, 4] | 16 | 666 |

Table 7: Architecture details of MCMAE small, base, large and huge. MCMAE-B* represents multi-scale encoder with large mlp-ratios in stage 1 and stage 2. $[C_1, C_2, C_3]$, $[L_1, L_2, L_3]$, $[E_1, E_2, E_3]$ and $[P_1, P_2, P_3]$ represents channel dimension, number of layer, spatial resolution and mlp-ratios for each stage 1, stage 2 and stage 3. $H_a$ stands for the number of attention heads in stage 3.

**Model Scaling up and down.** We design MCMAE of different parameters scales to match those of MAE-small, MAE-base, MAE-large and MAE-huge. Detailed network architectures are in appendix. The finetuning performances are shown in Table 8. Compared with the original MAE [28] of different scales, our MCMAE of different scales consistently outperform its MAE counterparts on Imagenet finetuning. This suggests that MCMAE can be an efficient learner for different paramter scales.

**Feature Map Visualization.** We provide some visualization of multi-scale feature maps generated by MAE and MCMAE backbone with the Mask R-CNN [30] method in Fig. 5. The masked convolution reveals much more fine-grained features compared with the pure vision transformer architecture of MAE, especially in feature maps with a stride of 4.

| Method | P-Epochs | Model size | | | | |
|--------|----------|-------|------|-------|-------|------|
| | | Small | Base | Base* | Large | Huge |
| MAE | 1600 | 79.5 | 83.6 | N/A | 85.9 | 86.9 |
| MCMAE | 800 | 82.6 | 84.6 | 84.9 | 86.2 | N/A |

Table 8: Ablation study of model scales.

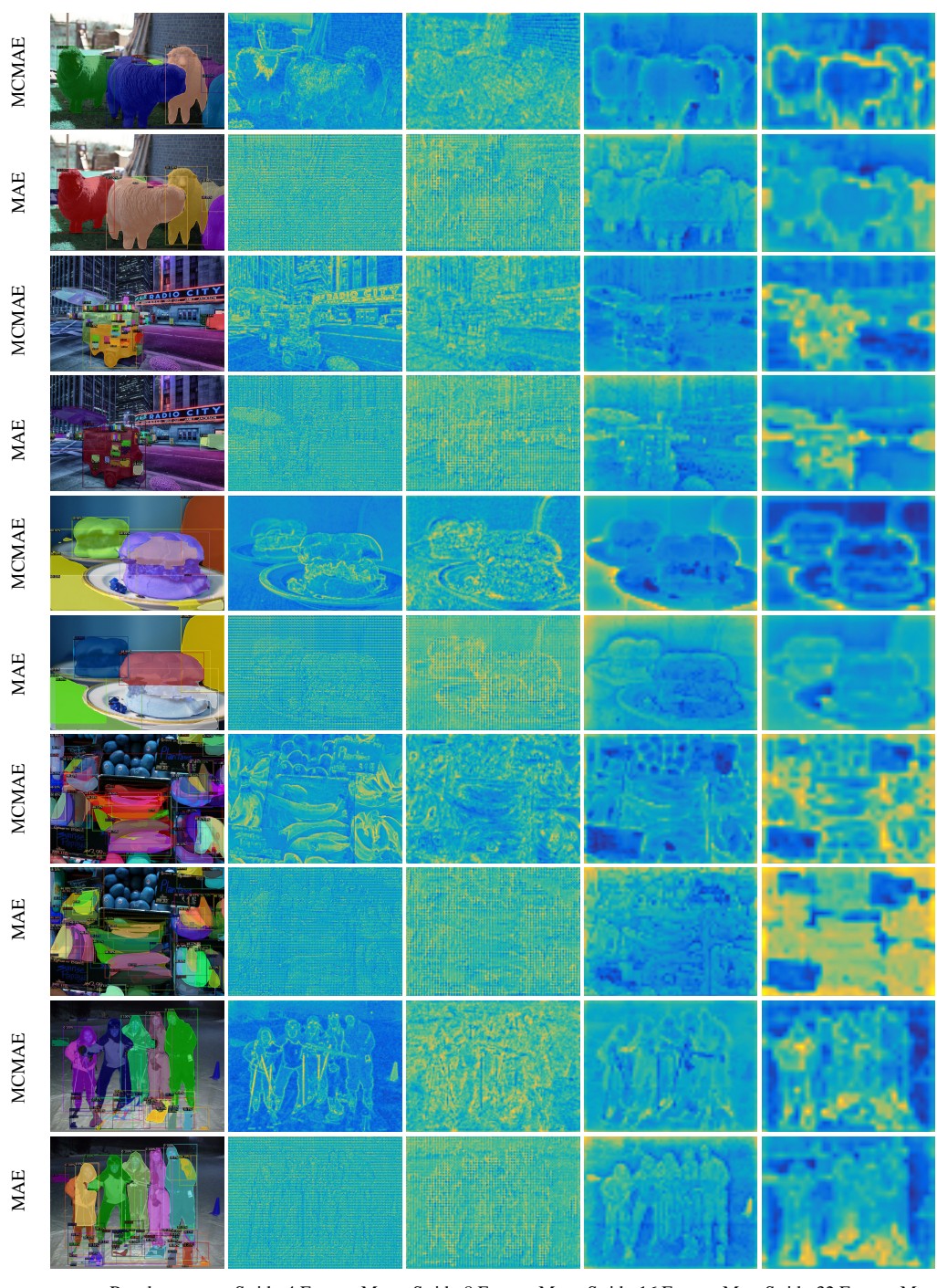

Figure 5: Visualization of feature maps with different strides.

