# OpenReview forum: "MCMAE: Masked Convolution Meets Masked Autoencoders"
_NeurIPS.cc/2022/Conference — NeurIPS 2022 Accept_

### Official Review · Reviewer_fnaM · 2022-07-01

**Rating:** 5
**Confidence:** 3
**Soundness:** 2 fair
**Presentation:** 3 good
**Contribution:** 3 good

**Summary:**

The paper starts with the hypothesis that a multiscale hybrid convolution-transformer can learn better representations using masked inputs than vanilla ViTs. The original masking scheme proposed in the MAE paper can be computationally prohibitive when directly applied to hybrid models. This paper presents a multiscale block-wise masking strategy with masked convolutions to efficiently train a hybrid transformer-convolutional model for representation learning. The paper shared a broad range of empirical results on classification, detection, segmentation, and video understanding tasks to show the effectiveness of the proposed technique.

**Questions:**

1. Did you experiment with larger models of the order of ViT-L and ViT-H? How do these results scale?
2. Why did you generate limited data points for VideoConvMAE and VideoConvMAE-multiscale in figure 3?
3. Why did you not include the CoAtNet results in the experiments?

**Limitations:**

Yes, authors adequately addressed the limitations.


**Strengths And Weaknesses:**

Originality: The novelty of the paper lies in its proposed multi-scale hybrid convolution-transformer encoder, which can generate hierarchical representations and possibly exceed the performance of vanilla ViTs. The idea of hybrid models already exists in multiple pieces of literature (CoAtNet, Early Convolutions etc.). Masked convolutions were introduced in the PixelRNN paper (https://arxiv.org/pdf/1601.06759.pdf). The strength of this paper is in its novel combination of existing ideas to produce a very simple hybrid framework that effectively combines the strength of convolutions and transformers.

I also like the idea of performing masking at the late stage and then progressively upsampling the mask to larger resolutions to avoid the requirement of keeping all tokens in stage 3.

The proposed setup naturally generates hierarchical representations and fits nicely with Feature Pyramid Networks. It is a nice way to generate a feature pyramid with local context via convolutions and global context using transformers.

Quality: The paper primarily describes experiments using ViT-B scale networks. It covers a broad set of vision tasks but it does not cover scale. It would be nice to see whether the proposed scheme continues to outperform existing masking techniques for larger models. There is also limited runtime comparison with existing techniques.

The paper shares very informative results of ablation experiments comparing random masking, regular convolutions, multi-scale decoders etc.

Clarity: The paper is very well written, with a nice flow, and explains the concepts with ease.
Nit: line 56 pretraing -> pretraining

Significance: The paper proposes a simple and effective hybrid convolution-transformer encoder, which naturally generates hierarchical representations from an image and outperforms a number of existing techniques.

---

> ### Author Response · Authors · 2022-08-01
> **Responses to Reviewer fnaM**
>
> > Did you experiment with larger models of the order of ViT-L and ViT-H? How do these results scale?
>
> We pretrain ConvMAE-Large with multi-scale fusion for 1600 epochs and fine-tune the pretrained ConvMAE-Large on ImageNet, COCO, and ADE20K. We compare the results with those of ConvMAE-Base, MAE-Base, and MAE-Large in the following table. It shows that our proposed ConvMAE can further improve when scales up. In the future, we will conduct experiments to train ConvMAE-Huge.
>
> |               | COCO FT Epoch | COCO AP{Box} | ImageNet | ADE20K |
> |---------------|:---------------:|:--------------:|:----------:|:--------:|
> | MAE-Base      |      100      |     51.2     |   83.6   |  48.1  |
> | ConvMAE-Base  |       25      |     53.2     |   85.0   |  51.7  |
> | MAE-Large     |      100      |     54.6     |   85.9   |  53.6  |
> | ConvMAE-Large |       25      |     55.6     |   86.7   |  54.1  |
>
> > Limited runtime analysis with current approaches.
>
> We compare the inference speed under Mask-RCNN framework for object detection. Our ConvMAE significantly improves the accuracy of object detection with slightly increase of inference time. The inference speed is tested on the A100 GPU.
> |               | Inference Speed | COCO AP{Box}
> |---------------|-----------------|-----------------|
> | ConvMAE-Base  | 0.090 s/img     |  53.2 |
> | MAE-Base      | 0.083 s/img     | 51.2  |
>
>
> > More experiments on VideoConvMAE and VideoConvMAE-multiscale
>
> We additionally pretrain VideoConvMAE for 400 epochs and VideoConvMAE-multiscale for 1600 epochs. The performance for 400 epochs is updated in the following table. We will update the results of 1600 epochs after the longer training is finished.
>
> | ConvMAE/Epochs | 200  | 400 | 800  |
> |----------------|------|-----|------|
> | Kinetics-400   | 80.1 |81.3| 81.7 |
> | SSV2           | 67.7 |69.2| 69.9 |
>
>
> | ConvMAE-multiscale/Epochs | 800  | 1600 |
> |----------------|------|-----|
> | Kinetics-400   | 82.7 |N/A|
> | SSV2           | 70.7 | N/A|
>
>
>
>
> > Performance comparison with CoAtNet
>
> |              | Parameters | Resolution | ImageNet |
> |--------------|------------|------------|----------|
> | CoAtNet-3      | 168 M      | 224 * 224  |   84.5   |
> | ConvMAE-Base | 88 M       | 224 * 224  |   85.0   |
>
> CoAtNet only performs experiments on image classification. The ImageNet-1K accuracy of CoAtNet and ConvMAE is listed above. ConvMAE with 88 million parameters can surpass CoAtNet-3 with 168 million parameters by 0.5 accuracy.

---

> > ### Author Response · Authors · 2022-08-09
> > **Update of VideoConvMAE-multiscale pretrained for 1600 epochs on SSV2**
> >
> > We update the results of VideoConvMAE-multiscale pretrained for 1600 epochs on SSV2 in the table below :
> > | ConvMAE-multiscale/Epochs | 800  | 1600 |
> > |----------------|------|-----|
> > | Kinetics-400   | 82.7 |N/A|
> > | SSV2           | 70.7 | 71.2|

---

### Official Review · Reviewer_qdV5 · 2022-07-09

**Rating:** 8
**Confidence:** 4
**Soundness:** 4 excellent
**Presentation:** 4 excellent
**Contribution:** 3 good

**Summary:**

This paper addresses the difficulty of applying MAE training with convolutional layers. The proposed ConvMAE adopts masked convolutions in the early stage of convolutional layers by applying convolution on the masked featuremaps. In this way, the information leak is prevented. With the proposed ConvMAE training, the ViT with early convolutional layers can benefit from the MAE training and achieved better transfer learning results comparing to standard ViT.  It achieves superior performance on ImageNet & MS-COCO.

**Questions:**

Would the masked featuremap introduce some bias or artifacts to the training along the mask edges? It may be helpful to find a way to alleviate the boundary issue.

**Ethics Review Area:**

["I don’t know"]

**Limitations:**

The comparison with the MAE training with standard ViT backbones may be unfair, due to introducing extra computation cost with the convolutional layers. It would be helpful to further break down the improvements.

**Strengths And Weaknesses:**

1. Novel training strategy to enable MAE training for models with convolutional layers.
2. Strong performance on various transfer learning tasks.

---

> ### Author Response · Authors · 2022-08-01
> **Responses to Reviewer qdV5**
>
> > Would the masked feature map introduce some bias or artifacts to the training along the mask edges? It may be helpful to find a way to alleviate the boundary issue.
>
> Yes. Masked feature map would introduce artifacts to the training process. We will explore new approaches for solving the artifacts introduced by masked feature.
>
> > The comparison with the MAE training with standard ViT backbones may be unfair, due to introducing extra computation cost with the convolutional layers. It would be helpful to further break down the improvements.
>
> ConvMAE shares similar parameters and FLOPs with MAE on various downstream tasks as shown in the table below (Table 1, 2, 3 of the original paper). When we design ConvMAE, we compensate for the extra computation of convolution layers by reducing the number of transformer blocks from 12 blocks to 11 blocks.
>
>
> | Method       | FLOPs/Params of SEG | FLOPs/Params of DET |
> |:--------------:|:---------------------:|:---------------------:|
> | MAE-Base     | 0.6 T / 163M        | 0.8T / 111M         |
> | ConvMAE-Base | 0.6 T / 153M        | 0.9T / 104M         |

---

### Official Review · Reviewer_mqzA · 2022-07-10

**Rating:** 6
**Confidence:** 3
**Soundness:** 3 good
**Presentation:** 2 fair
**Contribution:** 3 good

**Summary:**

This paper proposed a new self-supervised learning framework by integrating hybrid convolution-transformer architectures and masked convolution into the masked auto-encoders. The proposed method can achieve computational efficiency and low pretraining-finetuning gap at the same. Extensive experiments on several computer vision tasks demonstrate the effectiveness of the proposed method.

**Questions:**

See "Weaknesses".

**Limitations:**

Yes.

**Strengths And Weaknesses:**

__Strengths__

- The paper is well written and easy to follow. Sufficient technique details are provided.
- The proposed method is well motivated and simple. Several key components are proposed to address heavy computational cost and pretraining-finetuning discrepancy.
- The proposed method is flexible and can be applied in both image classification and object detection.

__Weaknesses__

- It seems hybrid convolution-transformer architectures have been explored in previous works but show how very similar performance to MAE (Lines 45-47). Why the proposed method can make them work for MAE? The differences from previous work and the contribution of the paper remains vague.
- Some parts of the method are not clearly illustrated. For example, in “Block-wise Masking with Masked Convolutions”, the authors state that “Uniformly masking stage-1 input tokens would cause all tokens of stage-3 to have partially visible information and requires keeping all stage-3 tokens”. Why the proposed method can address this issue? What is the key idea of the proposed method?
- The required training epochs vary from different methods. I wonder whether the proposed method can still outperform others under the same training epochs.

__Post Rebuttal__

I thank the authors for their response. Most of my concerns have been addressed. I increased my rating and recommend acceptance for this paper.

---

> ### Author Response · Authors · 2022-08-01
> **Responses to Reviewer mqzA**
>
> > The contributions of ConvMAE compared with previous hybrid architecture is vague?
>
> The overall architecture of ConvMAE shares similarity with previous efforts for hybrid visual backbone [1,2,3]. However, as we stated in Line58-72, the contribution of ConvMAE focuses on how to effectively and efficiently pretrain hybrid backbones under masked auto encoding settings with the following strategies:
> 1. The adopted masked convolution in the early stages prevents information leakage introduced by local convolutional operators.
> 2. The proposed blockwise masking strategy can avoid the requirement of keeping all tokens in stage 3, which significantly accelerates the pretraining.
> 3. The proposed multi-scale fusion decoder takes advantage of supervision signals for both fine-grained and coarse-grained features to learn more discriminative representations.
>
> As shown in Table 5 and Table 6, our proposed training strategies can effectively improve the representations of hybrid architecture  under MIM pretraining paradigm while maintaining the training efficiency of original MAE.
>
> [1] Early convolutions help transformers see better. \
> [2] CoAtnet: Marrying convolution and attention for all data sizes. \
> [3] Container: Context aggregation network.
>
>
>
> > Why “Block-wise Masking with Masked Convolutions” can keep 25% of tokens inside transformer block?
>
> Blockwise Masking strategy generates a mask for stage 3 then progressively upsamples it to obtain higher-resolution masks for stages 1 and 2. No information would be leaked to unmasked tokens of stage 3 if the corresponding upsampled masks are used for early stages. This strategy can make sure only 25% of all tokens need to be processed during transformer blocks.  On the contrary, uniformly masking input tokens requires keeping all tokens in stage 3.
>
> > The pretraining epochs of ConvMAE vary from other approaches. Can ConvMAE outperforms other approaches under a fair pretraining epochs?
>
> | Method   | Encoder | PT-Epochs | ImageNet-1K |
> |:----------:|:-----------:|:-----------:|:-------------:|
> | BEiT     | 100%|300       | 83.0        |
> | MAE      | 25%|1600      | 83.6        |
> | SimMIM   | 100%|800       | 84.0        |
> | MaskFeat | 100%|300       | 83.6        |
> | data2vec | 100%|800       | 84.2        |
> | ConvMAE  | 25%|200       | 84.4        |
>
>
> In the above table (Table 6 in our original submission), we present the ImageNet fine-tuning performance of ConvMAE pretrained for 200 epochs. Compared with previous approaches with longer pretraining epochs, ConvMAE can surpass them with a shorter training epoch by only processing 25% tokens inside encoder.

---

### Official Review · Reviewer_KmDT · 2022-07-11

**Rating:** 8
**Confidence:** 5
**Soundness:** 4 excellent
**Presentation:** 4 excellent
**Contribution:** 3 good

**Summary:**

This paper proposes a self-supervised framework using a hybrid convolution-transformer architecture, to obtain multi-scale, hierarchical representations. Masked convolution is introduced to prevent information leakage in convolution blocks and block-wise masking strategy is applied to improve computational efficiency. The resulting model achieves competitive performances in image classification and dense prediction tasks such as object detection.

**Questions:**

How would the duration of the pre-training epochs impact the backbone performances? How much performance drop if changing to shorter training epochs, such as reducing it from 1600 to 800 epochs?

**Limitations:**

adequate

**Strengths And Weaknesses:**

strengths:
1. This work effectively extends the self-supervised MAE framework to the hierarchical, convolution-transformer hybrid architecture.
2. The resulting model outperforms existing self-supervised models in classification and dense prediction tasks.

---

> ### Author Response · Authors · 2022-08-01
> **Responses to Reviewer KmDT**
>
> >How would the duration of the pre-training epochs impact the backbone performances? How much performance drop if changing to shorter training epochs, such as reducing it from 1600 to 800 epochs?
>
> | Pretrain Epochs | ImageNet FT | COCO AP box | ADE 20K |
> |:-----------------:|:-------------:|:-------------:|:---------:|
> | 200             | 84.1        | 50.2        | 48.1    |
> | 400             | 84.4        | 51.4        | 49.5    |
> | 800             | 84.6        | 52.0        | 50.2    |
> | 1600            | 84.6        | 52.5        | 50.7    |
>
> In the above table (Table 4 in our original submission), we study the influence of different pretraining epochs on various downstream tasks. Longer pretraining epochs lead to improved performance on detection and segmentation. The improvement on classification tasks saturates at 800 epochs. Please refer to above table (Table 4 in our original submission) for the performance comparison between model pretrained for 800 and 1600 epochs.

---

### Official Review · Reviewer_Nkn5 · 2022-07-12

**Rating:** 6
**Confidence:** 4
**Soundness:** 3 good
**Presentation:** 2 fair
**Contribution:** 3 good

**Summary:**

The paper proposes ConvMAE, a hybrid convolution-transformer architecture that is friendly to MAE-like pre-training. MAE was originally proposed with ViT, and due to omitted mask tokens in the backbone encoder, MAE is not trivially extensible to convolutional networks. The work extends MAE by resorting to the hybrid design of first using convolutions, and then using transformers. The masking is done block-wise (at the resolution of the transformer); and masked convolutions are used to avoid potential cheating. Extensive experiments are done on ImageNet classification, object detection, semantic segmentation, video classification. Various ablation analysis is also provided.

**Questions:**

- Please address the concerns mentioned above, especially the first point.

**Limitations:**

I do not see potential negative societal impact concern. The paper also points this out at the end, which is adequate to me.

**Strengths And Weaknesses:**

(+) Self-supervised learning, especially masked auto-encoding for images is an emerging topic in computer vision. A breakthrough in this direction can bear huge significance. The work aims at fixing the limitation of MAE by introducing an hybrid architecture of convolutions and transformers, which is definitely important and relevant to the NeurIPS audience.

(+) The paper is well written, and is clear enough for readers to follow through with good illustrations.

(+) The experiments are extensive and conclusive. The downstream transfers include image classification, object detection, semantic segmentation, and even video understanding is involved (which by itself could be an independent investigation). The ablations and the conclusions are also covering most of the things I can think of -- a solid paper clearly with a lot of hard work behind the scene.

(-) I think the "Conv" part of "ConvMAE" is an over-emphasis. The architecture only has 4 conv layers in the bottom of the network, while it has 11 transformer blocks for the base model (ViT-B has 12 blocks in total). So my current understanding is that ConvMAE has a similar architecture as in:

Xiao, Tete, et al. "Early convolutions help transformers see better." Advances in Neural Information Processing Systems 34 (2021): 30392-30400.

This means the majority of the architecture is still transformers, and in this regard, the difference/significance over original MAE is not that salient. This is the biggest concern about the paper -- it has a risk of over-sell with the term "Conv" in it.

(-) One minor concern is about the scalability of ConvMAE. The paper is highly focused on the model size of the base model. It is unclear the benefit of ConvMAE can still hold when the model size further scales up, as shown in pure ViT-based MAE.

(-) Some minor typos need to be fixed with proof-reading:, e.g., should define what is FOV at page 2, and the mask ratio should be 75% instead of 25% for MAE if I recall correctly.

---

> ### Author Response · Authors · 2022-08-01
> **Responses to Reviewer Nkn5**
>
> >ConvMAE appends several convolutional blocks upon transformer blocks. The difference between MAE and Early Conv is not that salient.
>
> The overall architecture of ConvMAE shares similarity with previous efforts for hybrid visual backbone [1,2,3]. However, as we stated in Line58-72, the contribution of ConvMAE focuses on how to effectively and efficiently pretrain hybrid backbones under masked auto encoding settings with the following strategies:
> 1. The adopted masked convolution in the early stages prevents information leakage introduced by local convolutional operators.
> 2. The proposed blockwise masking strategy can avoid the requirement of keeping all tokens in stage 3, which significantly accelerates the pretraining.
> 3. The proposed multi-scale fusion decoder takes advantage of supervision signals for both fine-grained and coarse-grained features to learn more discriminative representations.
>
> As shown in Table 5 and Table 6, our proposed training strategies can effectively improve the representations of hybrid architecture  under MIM pretraining paradigm while maintaining the training efficiency of original MAE.
>
> |               | COCO FT Epoch | COCO AP{Box} | ImageNet | ADE20K |
> |---------------|:---------------:|:--------------:|:----------:|:--------:|
> | MAE-Base      |      100      |     51.2     |   83.6   |  48.1  |
> | ConvMAE-Base  |       25      |     53.2     |   85.0   |  51.7  |
>
> As shown in above table (Table 1, 2, 3 in our submission), ConvMAE significantly outperforms MAE with simple but effectively strategies.
>
>
> [1] Early convolutions help transformers see better. \
> [2] CoAtnet: Marrying convolution and attention for all data sizes. \
> [3] Container: Context aggregation network.
>
> > It is unclear the benefit of ConvMAE can still hold when the model size further scales up, as shown in pure ViT-based MAE.
>
> We pretrain ConvMAE-large with multi-scale fusion for 1600 epochs and fine-tune on ImageNet, COCO, and ADE20K. We compare with ConvMAE-Base, MAE-Base, and MAE-Large in the following table. The further scaling-up experiments verify the scaling capability of ConvMAE.
>
> |               | COCO FT Epoch | COCO AP{Box} | ImageNet | ADE20K |
> |---------------|:---------------:|:--------------:|:----------:|:--------:|
> | MAE-Base      |      100      |     51.2     |   83.6   |  48.1  |
> | ConvMAE-Base  |       25      |     53.2     |   85.0   |  51.7  |
> | MAE-Large     |      100      |     54.6     |   85.9   |  53.6  |
> | ConvMAE-Large |       25      |     55.6     |   86.7   |  54.1  |
>
> >Minor typos.
>
> Thanks for pointing out the typos. FOV stands for field-of-view. The mask ratio shall be 75% instead of 25%. We will modify those typos in the updated version.

---

> > ### Comment · Reviewer_Nkn5 · 2022-08-04
> > **Over-emphasis issue of the name "ConvMAE" is still not addressed well**
> >
> > Thanks for the response and glad to see this approach improving over the original MAE. However, my biggest concern that the over-emphasis on the name is not addressed. Several additional remarks:
> >
> > * "ConvMAE" gives reader the wrong impression that it is a ConvNet architecture with MAE pre-training. It is not the case here (hybrid architecture used).
> > * As mentioned by the authors, none of the 3 works [1-3] the paper followed, or has similar architecture with, use the name "Conv" in their approach. For [1], it is citing the architecture as ViT-C (and instead of ViT-B), where Transformer is still the main character in play.
> > * May be a better name is MCAE, as the paper is Masked Convolutions meets masked Autoencoders?
> >
> > If the biggest concern is not addressed, I will lower my rating of the paper for the potential risk of oversell.

---

> > > ### Author Response · Authors · 2022-08-04
> > > **Responses to Reviewer Nkn5**
> > >
> > > Thanks for your suggestion. After careful consideration, we will rename our approaches to MC-MAE (Masked Convolution meets Masked AutoEncoder) and backbone ConViT to CViT. The final version will be updated with the new name.

---

> > > > ### Comment · Reviewer_Nkn5 · 2022-08-05
> > > > **Thanks!**
> > > >
> > > > Thanks! Given this change I have no other concerns about the paper.

---

### Meta-Review · Area_Chair_EZ3D · 2022-08-25

**Recommendation:** Accept
**Confidence:** Certain

**Metareview:**

The reviewers are positive about this submission initially. After the authors' rebuttal, one reviewer pointed out that the name `ConvMAE' is not proper to describe the current work. The authors respond by claiming using an alternative name, which is acknowledged by the reviewer. Overall, all the reviewers stand positive for this work and AC stands with the reviewers. The authors shall take the suggestions from the reviewers to further polish the current work in the camera-ready submissions.

**Award:**

No

---

### Decision · Program_Chairs · 2022-09-14

Accept